# Rapid Processing of In-Doped ZnO by Spray Pyrolysis from Environment-Friendly Precursor Solutions

Nina Winkler [1,2,*] , Adhi Rachmat Wibowo [1] , Bernhard Kubicek [1], Wolfgang Kautek [2] , Giovanni Ligorio [3] , Emil J. W. List-Kratochvil [3] and Theodoros Dimopoulos [1,*]

[1] AIT Austrian Institute of Technology, Center for Energy, Photovoltaic Systems, Giefinggasse 6, 1210 Vienna, Austria; Rachmat.Wibowo@ait.ac.at (A.R.W.); Bernhard.Kubicek@ait.ac.at (B.K.)

[2] Department of Physical Chemistry, University of Vienna, Währinger Straße 42, A-1090 Vienna, Austria; wolfgang.kautek@univie.ac.at

[3] Institut für Physik, Institut für Chemie, and IRIS Adlershof, Humboldt-Universität zu Berlin, Brook-Taylor-Str. 6, 12489 Berlin, Germany; giovanni.ligorio@hu-berlin.de (G.L.); emil.list-kratochvil@hu-berlin.de (E.J.W.L.-K.)

\* Correspondence: nina.winkler@ait.ac.at or a01026695@unet.univie.ac.at (N.W.); theodoros.dimopoulos@ait.ac.at (T.D.)

**Abstract:** This study focused on the deposition of indium-doped zinc oxide (IZO) films at high growth rates by ultrasonic spray pyrolysis. We investigated the influence of processing parameters, such as temperature and solution flow rate, on the structural, optical, and electrical film properties. For all depositions, low-cost and low-toxicity aqueous solutions and metal salt precursors were used. Through the optimization of the spraying parameters and pattern, a spatially homogeneous IZO layer with transparency greater than 80%, resistivity of $3.82 \times 10^{-3}$ $\Omega \cdot$cm for a thickness of 1800 nm (sheet resistance of 21.2 $\Omega$/sq), Hall carrier density of $1.36 \times 10^{20}$ cm$^{-3}$, Hall mobility of 12.01 cm$^2$ V$^{-1}$ s$^{-1}$, and work function of 4.4 eV was obtained. These films are suitable for implementation in optoelectronic and photovoltaic devices.

**Keywords:** transparent conductive electrode; TCO; zinc oxide; indium-doped ZnO; thin film; oxide semiconductor; spray pyrolysis; aqueous solution deposition

## 1. Introduction

Transparent conducting oxides (TCOs) are important materials for various applications, such as flat-panel displays [1], functional windows [2], light-emitting diodes [3], and photovoltaics [4]. Established TCOs are wide-band gap (>3 eV), *n*-type semiconductors based on indium oxide (In$_2$O$_3$), tin oxide (SnO$_2$), or zinc oxide (ZnO) that have been doped with donors to increase the charge carrier concentration. ZnO-based TCOs are especially attractive due to their low cost and ease of fabrication through solution-based deposition techniques. To attain a high carrier concentration and improve the film conductivity, ZnO can be doped with the group III elements Al, Ga, or In. Depending on the deposition methods and conditions, different dopants result in distinctive film properties. For spray pyrolysis—a versatile and low-cost solution-based deposition technique—it was shown that doping with In promotes lower film resistivity compared to Al or Ga, even at low In concentrations [5–7].

Previously, we reported on the spray pyrolysis of In-doped ZnO (IZO) films with Zn-acetate and In-acetylacetonate (In(acac)$_3$) as precursors [8]. Although the films were highly transparent and conductive, the film growth rate was restricted to ~10 nm/min. To obtain films with sufficiently low resistivity for device applications, the deposition time had to be extended to several hours. Further, due

to the low solubility of In(acac)$_3$ in water, it was necessary to stir the solution at 80 °C for at least 2 h. This resulted in an extended fabrication time, which was disadvantageous with respect to upscaling. Rapid deposition would also reduce the amount of alkali ions diffusing from the glass substrate during spraying, which can act as acceptors in ZnO and decrease the conductivity [9].

Literature reports on the spray pyrolysis of IZO films demonstrated that it was challenging to obtain both high growth rates and high-quality films at the same time. Currently, rapidly processed films possess either poor electronic properties [10] or low transparency (~60%) [11]. Considering this, the main purpose of the present study was to increase the growth rate of IZO films on glass substrates while maintaining a high film transparency and low resistivity. This was accomplished by using In-acetate (InAc$_3$) as a precursor with a higher solubility compared to In(acac)$_3$ in aqueous solutions. Further, the influence of the main spraying parameters on the growth rate and film properties was investigated. Notably, InAc$_3$ also has a lower environmental and health impact compared to In(acac)$_3$ [12].

## 2. Materials and Methods

Borosilicate glass substrates (Schott Nexterion®D, Mainz, Germany. $7.5 \times 2.5$ cm$^2$) were prepared as previously reported [13]. In summary, they were ultrasonically cleaned in a Hellmanex®III (Munich, Germany) washing solution, rinsed with deionized water (DI, 18 MΩ·cm$^{-1}$) and isopropanol, and dried in nitrogen stream. The prepared optimized precursor solution contained 0.2 M zinc acetate dihydrate (ZnAc$_2 \times$ 2H$_2$O, Sigma-Aldrich 96459 (Munich, Germany)), 4 mol % indium acetate (InAc$_3$, Sigma-Aldrich 510270) dissolved in DI water with 8 vol % acetic acid (HAc, Sigma-Aldrich A6283). For complete dissolution of the precursor salts, the solution was placed for 10 min in an ultrasonic bath at 25 °C.

The ultrasonic spray pyrolysis was carried out on a Sono-Tek ExactaCoat®system (Milton, NY, USA), equipped with a 120 kHz Sono-Tek Impact®ultrasonic nozzle in horizontal geometry. The fine droplet mist was directed towards the hot plate which was covered with an alumina substrate holder using compressed air (0.5 bar). The temperature varied between 360 and 400 °C, due to the contribution of a volatile basic zinc acetate complex to the IZO film formation under this temperature regime [14]. The flow rate was varied between 0.8 and 2.4 mL/min to obtain a steady spraying cone and to avoid precipitates. The spraying volume was a constant 45 mL in all cases, except when using the optimized spraying pattern.

The deposited films were characterized by scanning electron microscopy at an accelerating voltage of 5 kV and in-lens detector (SEM, Zeiss, Ultra 40, Oberkochen, Baden-Württemberg, Germany), atomic force microscopy (AFM, Molecular Imaging, Pico Plus, San Diego, CA, USA) in tapping mode, X-ray diffraction using Cu Kα (λ = 1.5419 Å) radiation (XRD, ThermoFisher Scientific, ARL Equinox 100, Waltham, MA, USA), Fourier transform infrared spectroscopy (FTIR, Bruker Vertex 70, Billerica, MA, USA) and 4-point probe set-up connected to a semiconductor parameter analyzer (Süss MicroTec probes, Garching, Bayern, Germany and Agilent 4156 C parameter analyzer, Santa Clara, CA, USA). The film growth rate was extracted from SEM cross sections by taking the mean value of several film thickness measurements. The optimized IZO films were additionally characterized by Hall effect measurements in the van der Pauw geometry with a magnetic field of 0.27 T. Ultraviolet photoemission spectroscopy (UPS) was performed in a JEOL JPS-9030 photoelectron spectrometer system (Japan Electron Optics Laboratory JEOL, Akishima Tokyo, Japan) using a monochromatic E-LUX (Excitech Ltd., Enfield, Middlesex, UK) light source as excitation source (10.2 eV). The samples were electrically grounded during the measurements of the valence band, while a bias of −5 V was applied for determining the secondary electron cutoff (SECO).

## 3. Results and Discussion

The solution composition influences the properties of the films prepared by spray pyrolysis in various ways [15,16]. In the present study, the amount of zinc acetate was fixed to 0.2 M, and the amount of indium acetate varied between 3 and 5 mol %. It was observed that the solution containing

4 mol % InAc$_3$ resulted in a slightly lower film resistivity compared to samples deposited from 3 and 5 mol % InAc$_3$ solutions.

To investigate the influence of the acetic acid concentration on the film properties, 0–16 vol % of acetic acid (HAc) was added to the aqueous solution containing 4 mol % of InAc$_3$. The role of acetic acid was to stabilize the volatile indium- and zinc-acetate complexes in the solution and adjust the pH to its optimum value for the deposition [17]. A low HAc concentration resulted in unwanted precipitates on the film surface, while a high HAc concentration caused an increase in film resistivity. In accordance with our previous results [8], the lowest resistivity ($7.4 \times 10^{-3}$ $\Omega\cdot$cm) combined with highest transparency (>80%) were obtained for ~1500 nm-thick films deposited from a solution containing 8 vol % HAc (pH = 3.65).

### 3.1. Film Growth Rate

The film growth rate was influenced by the solution composition in various ways. It was shown that the type of precursor metal salt influenced the growth of ZnO film [18] and a pH value between 3.5 and 4.3 resulted in the highest ZnO growth rate, due to zinc acetate complexes forming at this pH range [19]. Within the narrow concentration range for InAc$_3$ (3–5 mol %) and HAc (4–16 vol %) in this study, there was practically no obvious influence on the growth rate, as shown in Figure 1a,b. In comparison to our previous study [8], changing the zinc acetate concentration from 0.1 to 0.2 M and the type of indium precursor salt resulted in a ~4-fold increase of the growth rate, as displayed in Figure 1. In Figure 1c it can be observed that increasing the deposition temperature from 360 to 400 °C resulted in a minor gradual decrease of the growth rate. A higher flow rate ($v$), as displayed in Figure 1d, offers the possibility to accelerate the growth process. With constant spraying volume and adjustment of the spraying cycles, the highest possible $v$ to avoid undesired salt precipitation, due to incomplete precursor decomposition, was 2.4 mL/min. Although the step from 0.8 to 1.6 mL/min led to a significant increase of growth rate, a further increase to 2.4 mL/min had practically no effect on the growth rate.

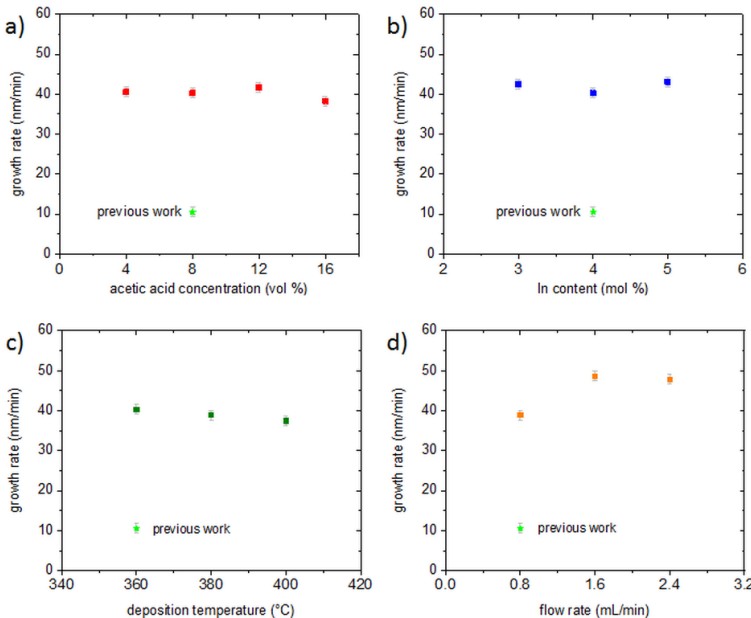

**Figure 1.** Film growth rate dependency on the (**a**) acetic acid concentration, (**b**) indium acetate concentration, (**c**) deposition temperature, and (**d**) flow rate compared to our previous results [8].

### 3.2. Influence of Deposition Temperature and Solution Flow Rate on Film Properties

While substrate temperature is a parameter frequently investigated in the spray pyrolysis of IZO films [20–23], there are only a few studies that examine the influence of the precursor solution flow rate on IZO film properties [21,24]. Further, none of the abovementioned reports use the same solvent and

precursor combination as the present study. Both the precursor type and the solvent composition have been shown to have a significant influence on film properties [17,25–27]. Table 1 lists film thickness, growth rate, film resistivity, and sheet resistance for varying substrate temperatures and solution flow rates. In this study, when the flow rate was doubled and the temperature was kept constant at 380 °C, the number of spraying cycles needed to be reduced by the same factor to obtain comparable film thicknesses (compare IZO-2 with IZO-4/IZO-5 in Table 1).

**Table 1.** Film properties for variations in deposition temperature and solution flow rate. IZO, indium-doped zinc oxide.

| Exp. No. | Temp. (°C) | Flow Rate (mL/min) | Spraying Cycles | Film Thickness (nm) | Growth Rate (nm/min) | Resistivity ($\Omega\cdot$cm) | Sheet Resistance ($\Omega$/sq) | Crystallite Size (nm) |
|---|---|---|---|---|---|---|---|---|
| IZO-1 | 360 | 0.8 | 540 | 1450 | 40.3 | $7.4 \times 10^{-3}$ | 51 | 12.1 |
| IZO-2 | 380 | 0.8 | 540 | 1400 | 38.9 | $7.3 \times 10^{-3}$ | 52 | 15.8 |
| IZO-3 | 400 | 0.8 | 540 | 1350 | 37.5 | $1.4 \times 10^{-2}$ | 104 | 16.1 |
| IZO-4 | 380 | 1.6 | 270 | 1750 | 49.8 | $4.7 \times 10^{-3}$ | 27 | 17.4 |
| IZO-5 | 380 | 2.4 | 180 | 1720 | 47.8 | $4.6 \times 10^{-3}$ | 27 | 16.7 |
| IZO-6 | 380 | 1.6 | 200* | 1500 | 53.3 | $5.7 \times 10^{-3}$ | 38 | 14.4 |

* lower number of spraying cycles and same conditions as IZO-4 to obtain a lower film thickness.

A deposition temperature of 400 °C resulted in a higher resistivity compared to 360 and 380 °C. The lowest resistivity was obtained with an increased $v$ of 1.6 and 2.4 mL/min. To allow for comparison of samples with a similar thickness, IZO-6 was deposited using the same conditions as for IZO-4, but with a lower number of spraying cycles. IZO-6 films showed a decreased resistivity compared to films deposited at the same temperature with a lower flow rate (IZO-2), and this was independent of the reduced thickness.

The SEM plane-view image in Figure 2a shows that a lower temperature and a $v$ of 0.8 mL/min resulted in a reduced grain size and more compact films. The most significant influence on film morphology was caused by raising the deposition temperature to 400 °C while maintaining a flow rate of 0.8 mL/min, as seen in Figure 2c. In this case, larger elongated grains or grain agglomerates were observed. Increasing the flow rate at a constant temperature of 380 °C resulted in the elongation of small-sized grains (compare Figure 2b,d). Figure 2f shows the cross section image of the IZO-4 film deposited at 380 °C with a flow rate of 1.6 mL/min.

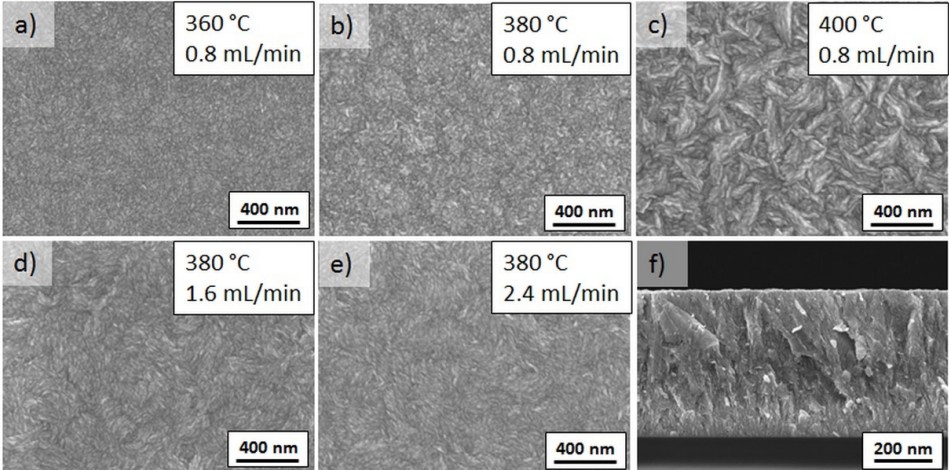

**Figure 2.** SEM plane views and cross section of (**a**) IZO-1; (**b**) IZO-2; (**c**) IZO-3; (**d**) IZO-4; (**e**) IZO-5; and (**f**) IZO-4.

The roughness of the films was also studied by AFM, as shown in Figure 3a–e. In accordance with observations from SEM images, the film deposited at low temperature and flow rate showed the

lowest film roughness with a root mean square (RMS) of 6.4 nm. Increasing the deposition temperature and the flow rate tended to increase the surface roughness, with the highest RMS value obtained for the film deposited at 380 °C and 2.4 mL/min flow rate.

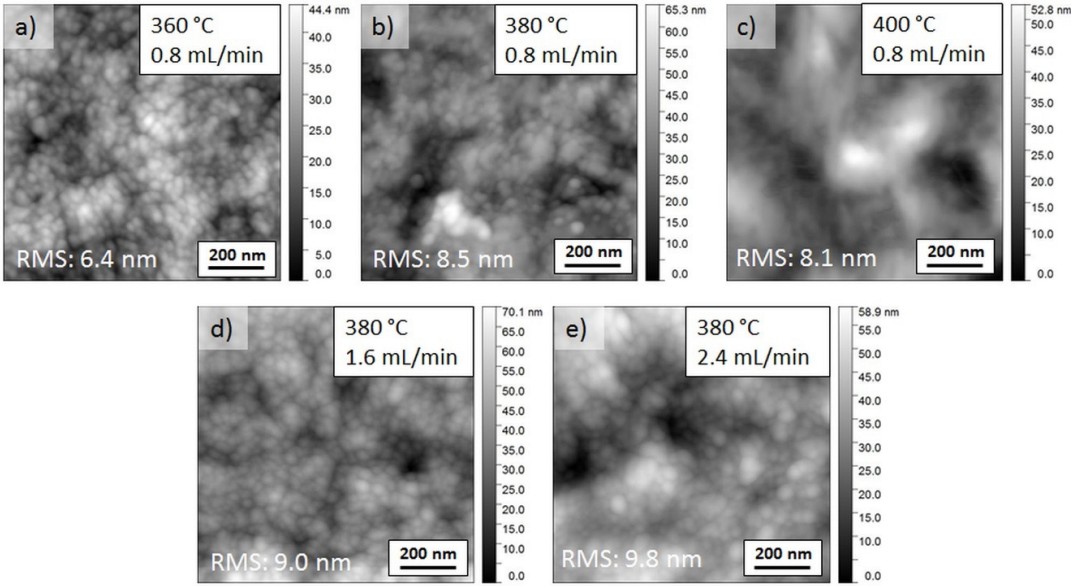

**Figure 3.** AFM images of (**a**) IZO-1; (**b**) IZO-2; (**c**) IZO-3; (**d**) IZO-2; (**e**) IZO-4; and (**f**) IZO-5.

X-ray diffractograms, as seen in Figure 4, showed that all films have a polycrystalline ZnO wurtzite crystal structure (COD: 96-900-4182), showing a high-intensity (10$\bar{1}$1) reflection and a lower-intensity (11$\bar{2}$0) reflection independent of the film thickness (compare IZO-4 and IZO-6). The crystallite sizes were calculated using the Scherrer equation [28] and are listed for all samples in Table 1. It was observed that there was a minor crystallite size increase from 12.1 to 16.1 nm with higher temperature, even though the film thickness decreased with temperature. This was in accordance with the observations from SEM images. Thick films deposited with high flow rates (IZO-4 and IZO-5) showed the largest crystallite sizes of 17.4 and 16.7 nm. IZO-6 films exhibited a smaller crystallite size (14.4 nm) that was similar to films deposited at the lower flow rate.

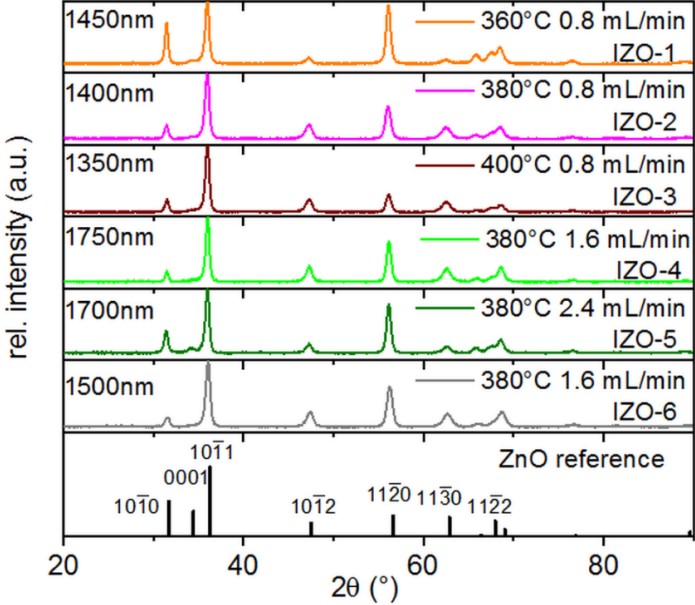

**Figure 4.** X-ray diffractograms for IZO films deposited with various temperatures/flow rates.

Transmittance spectra were measured between 300 and 2500 nm for different deposition temperatures, as seen in Figure 5a,b. All films exhibited Fabry–Pérot interferences that are commonly observed in compact thin films. With increasing temperature, the transmittance was enhanced. The highest absorption in the infrared region, which could be attributed to a higher amount of free carriers, was observed at 360 °C. The band gap was extracted using the Tauc formula [29] and can be observed in the inset of Figure 5a. The maximum value was 3.23 eV for 360 °C, decreasing to 3.14 and 3.17 eV, for 380 and 400 °C, respectively. The band edge shift to lower wavelengths was in agreement with the more pronounced free carrier absorption and may be explained by the Moss-Burstein effect due to a higher carrier concentration [30].

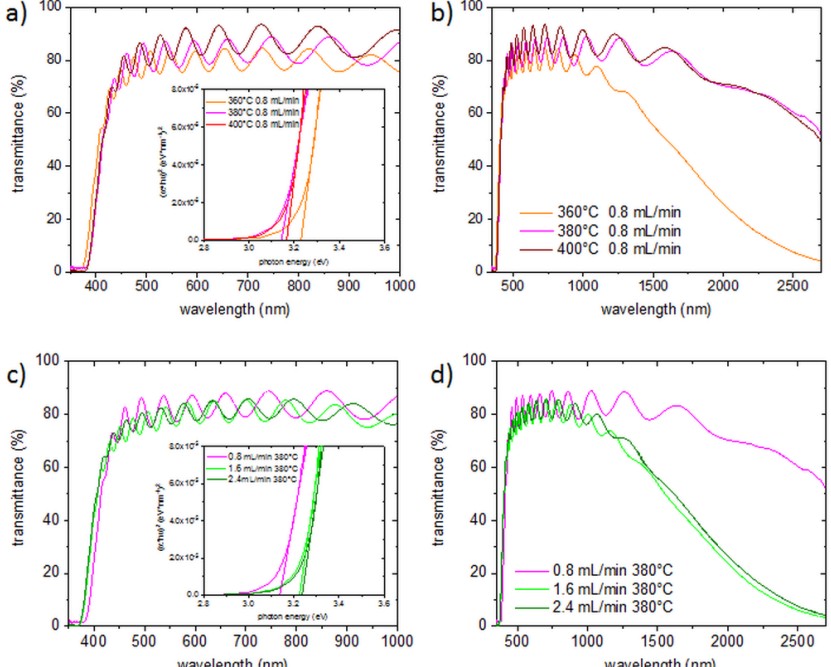

**Figure 5.** FTIR spectra for (**a**,**b**) variations in the deposition temperature and (**c**,**d**) variations in the solution flow rate. Insets in (**a**,**c**) show the Tauc plots for extraction of the optical band gap.

Transmittance spectra for films deposited with different solution flow rates are shown in Figure 5c,d. Higher flow rates caused only a minor decrease in transmittance for wavelengths up to 1000 nm, but a pronounced absorption in the infrared region, which could again be attributed to a higher amount of free carrier present in these films. The higher carrier concentration with increasing solution flow rate may have been due to a superior amount of substitutionally incorporated $In^{3+}$ in the film. Also, the lower effective substrate temperature that may arise from the higher flow rate (more pronounced cooling of the substrate with increased flux of material) could have been another factor. A band gap of 3.23 eV was observed for samples deposited with 1.6 or 2.4 mL/min, which was higher than the that for the film deposited at 0.8 mL/min. This was in accordance with the tendancy observed in the IR part of the spectrum. Further, it was seen that for $v >$ 2.4 mL/min, incomplete precursor deposition resulted in a higher amount of precipitates on the film surface, leading to less conductive films with lower transmittance.

### 3.3. Optimized Spraying Pattern

To further accelerate the deposition process and decrease the variations in film thickness over the 2.5 × 2.5 cm$^2$ glass substrate, an improved spraying pattern was introduced, as shown in Figure 6. The new spraying pattern increased the material influx. Due to this, it was necessary to decrease the flow rate from 1.6 to 1.2 mL/min to avoid substantial substrate cooling, which resulted in precipitation

on the substrate surface. The flow rate of 1.2 mL/min gave rise to a film growth rate of 55.4 nm/min (compare to Figure 1). For a 1800 nm-thick film, a sheet resistance of 21.2 Ω/sq was obtained and the same transparency of >80% was maintained. The faster film growth with the new spraying pattern can be explained by the reduced loss of volatile metal salt precursors to the escaping gas phase, which instead contribute to film formation. This was also demonstrated by the lower amount of precursor solution needed (36 instead of 45 mL) as in the previous experiments for approximately the same film thickness, rendering the film deposition process more efficient. For films deposited with these optimized deposition conditions, Hall measurements in the van der Pauw geometry were conducted, and a Hall carrier density of $1.36 \times 10^{20}$ cm$^{-3}$, a Hall mobility of 12.01 cm$^2 \cdot$V$^{-1} \cdot$s$^{-1}$, and a film resistivity of $3.82 \times 10^{-3}$ Ω·cm were determined. An image of the IZO film on glass is shown in Figure 6c. For the implementation of IZO electrode in the devices, the electronic band structure is crucial. Therefore, the work function, φ, and the onset of the valence band (VB) for the optimized IZO film were determined by UPS measurements. The schematic band diagram and the related measurements are shown in Figure 7a–c. The obtained VB onset was 3.2 eV and the work function was 4.4 eV.

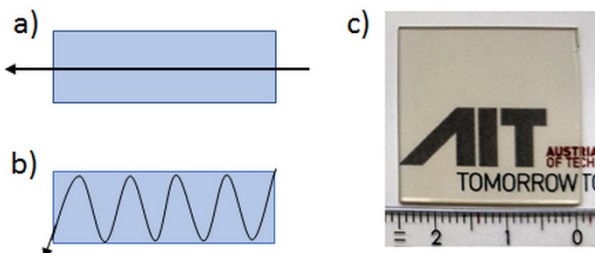

**Figure 6.** (**a**) Spraying pattern used in first experiments, (**b**) optimized spraying pattern, and (**c**) image of glass/IZO films with the optimized spraying protocol.

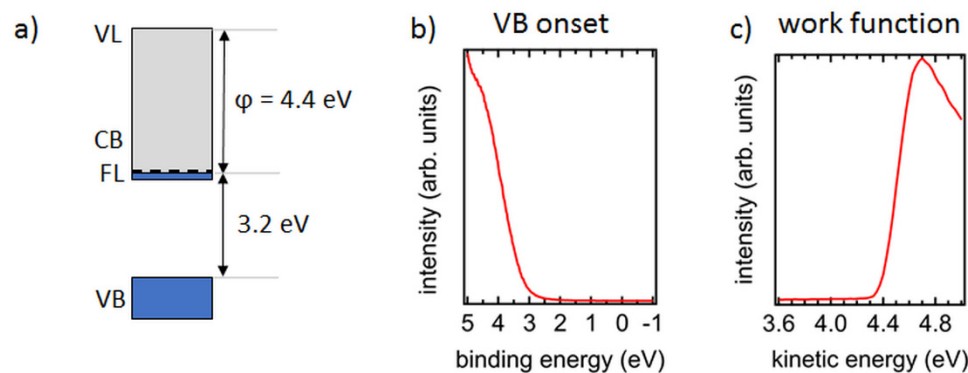

**Figure 7.** (**a**) Schematic IZO band diagram, (**b**) valence band onset, and (**c**) secondary electron cutoff (SECO) obtained from UPS measurements.

## 4. Conclusions

This study showed how highly transparent and conductive indium-doped ZnO electrodes can be fabricated from environment-friendly aqueous precursor solutions using a rapid spray pyrolysis process. It was observed that the film growth rate was substantially influenced by the type and concentration of the zinc precursor salt, while the amount of acetic acid and indium acetate showed a minor influence for the concentrations used in this study. The IZO film growth rate could further be accelerated by increasing the solution flow rate and optimizing the spraying pattern, resulting in less precursor species being lost to the gas phase during deposition. The optimized spraying process resulted in films with excellent electrical, structural, and optical properties, combined with high growth rates.

**Author Contributions:** Conceptualization, N.W. and T.D.; Investigation, N.W., G.L.; Resources, B.K, A.R.W., G.L., E.J.W.L.-K.; Writing—Original Draft Preparation, N.W. and T.D.; Writing—Review and Editing, N.W., W.K. and T.D.; Supervision, T.D. and W.K.

**Funding:** This research received no external funding.

**Acknowledgments:** The authors thank Stefan Edinger for valuable discussions.

**Conflicts of Interest:** The authors declare no conflict of interest.

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
