# Peer review of "Rapid Processing of In-Doped ZnO by Spray Pyrolysis from Environment-Friendly Precursor Solutions"

_coatings, doi:10.3390/coatings9040245_

Round 1
Reviewer 1 Report
In Table 1, IZO film properties for variations in deposition temperature and solution flow rate was modulated. Comparing IZO-2 and IZO-4, the film has similar thickness. Why the lowest resistivity was obtained with increased v of 1.6?
In figure 4, why is your measurement of FTIR so unstable? Is it a problem with the measuring machine or the film instability? Furthermore, why are the results of figure4 (b) and figure 4(d) different under the same process conditions( 0.8ml/min 380oC )?
In line 61-62 of second page, you changed that temperature and solution flow rate varied between 360 and 400°C and 0.8 and 2.4 ml/min, respectively. Is there any special significance in choosing this range?
In line 72-73 of second page, you stated that “The solution composition influences ‘in’ various ways”. The preposition should be modified with ‘on’.
In line 126-127 of fourth page, you claimed “Increasing the flow rate, at constant temperature of 380°C, has the effect that small-size grains become more elongated (compare figures 2(d), (e) and (f))”. However, the figure 2(e), and figure 2(f) looks similar, please show more high resolution below 400nm scale to verify elongated phenomenon.
In line 135-137 of fifth page, you reported “X-ray diffractograms in figure 3 show that all films have polycrystalline, ZnO wurtzite crystal structure, showing a high-intensity (101̅1) reflection and a lower intensity (112̅0) reflection, independent from the film thickness (compare IZO-4 and IZO-6). Then, the reference on your study shows ZnO XRD instead of “IZO” XRD, please correct it.
In line 137-143 of fifth page, you mentioned ,“The crystallite size was calculated by the Scherrer equation and is listed for all samples in table 1. It was observed that there is a minor crystallite size increase from 12.1 to 16.1 nm with higher temperature even though the film thickness decreased with the temperature. This is in accordance with the observations from SEM images. Thick films deposited with high flow rates (IZO-4 and IZO-5) show the largest crystallite sizes of 17.4 and 16.7 nm. IZO-6 films exhibit a smaller crystallite size (14.4 nm) that is similar to films deposited at the lower flow rate”, Why not put this text above Table 1, which strongly explains your description and s easier for the reader to understand.
Author Response
Dear reviewer,
the authors thank the reviewer for the evaluation of our manuscript and the helpful comments to improve the quality of this article. The revised article includes additional characterization (AFM, UPS) of the In-doped ZnO films deposited by spray pyrolysis. A point-by-point response to each comment is included as extra document and changes in the revised manuscript are highlighted in green. The authors hope that the revised version of the manuscript meets the requirements of the reviewers and the journal.
Yours sincerely,
Nina Winkler

Reviewer 2 Report
Comments to the Author
The manuscript presents highly transparent and conductive Indium-doped ZnO electrodes via a rapid spray pyrolysis process from environmental friendly aqueous precursors. Several key parameters such as temperature, flow rate and spraying cycles have been tuned to be well correlated with the optical and electronic properties. The established method of ultrasonic spray pyrolysis could provide important hints to deposit high quality of TCOs. I recommend publishing this work after authors consider below comments.
A couple of comments:
1. The roughness is very important criteria to assess the morphology and quality of TCOs. For example, in terms of conventional ITO and FTO, the roughness could determine the device structure and contact layers in the application of optoelectronics and PV devices. Author has mentioned roughness issue when mentioning the morphology influenced by temperature change, but not in detail. It would be nice to obtain roughness information via like AFM to be compared with each other.
2. The bandgap for variable IZOs has been clarified by Author, but there is no information for working function level. This level is quite crucial for TCOs to be combined with other layers, i.e. electron or hole transfer layers, for the proper design of energy level in device application. It would be nice if Author can provide this kind of information.
Some other minor comments:
1. Page 1 Line 32, “it was shown” could be replaced with “it showed” or “it has shown”.
2. Page 1 Line 39, “This results to” could be replaced with “This results in”.
3. Page 1 Line 46-48, “This was accomplished….and film properties”, a bit long sentence to be understood well, better to rephrase it.
4. Page 3 Line 109-110, “when the flow rate was doubled, the number…to obtain comparable film thickness”, as my understanding, you compare IZO-3 with IZO-4, the flow rate was doubled, but I note that the temp was also changed from 400C to 380C, another parameter changed, and how can you claim that the comparable film thickness was obtained only by reducing the spraying cycles? Please clarify.
5. Page 4 Line 123, “in compact films” delete “in”.
6. Page 6 Line 172, “precpitates” should be “precipitates”.
Author Response

(The authors gave the same response as above.)

Round 2
Reviewer 1 Report
The manuscript can be accepted from my point of view.